# The Effects of Caustic Soda and Benzocaine on Directed Grooming to the Eyestalk in the Glass Prawn, *Palaemon elegans*, Are Consistent with the Idea of Pain in Decapods

**DOI:** 10.3390/ani14030364

**Published:** 2024-01-23

**Authors:** Stuart Barr, Robert W. Elwood

**Affiliations:** School of Biological Sciences, Queen’s University, Belfast BT9 5DL, UK; s_barr@hotmail.com

**Keywords:** pain, decapod, caustic soda, benzocaine, eyestalk, *Palaemon elegans*

## Abstract

**Simple Summary:**

The possibility of pain occurring in animals is often accepted if various criteria are fulfilled. These criteria include prolonged grooming or rubbing at the site of a wound or tissue damage, or other behaviour involving the site of damage. We also expect to see a reduction in such activities if a local anaesthetic is applied. Here, we report on an experiment that applied caustic soda, a known irritant in humans, to one eyestalk of the glass prawn. This caused immediate escape responses and then nipping and picking at the treated eyestalk rather than at the untreated eyestalk. Prior application of a local anaesthetic reduced the amount of directed behaviour. However, the local anaesthetic also appeared to be an irritant as it too caused immediate escape responses and directed behaviour to the eyestalk. The results provide further support to the idea that these animals can experience pain.

**Abstract:**

Acceptance of the possibility of pain in animals usually requires that various criteria are fulfilled. One such criterion is that a noxious stimulus or wound would elicit directed rubbing or grooming at the site of the stimulus. There is also an expectation that local anaesthetics would reduce these responses to damage. These expectations have been fulfilled in decapod crustaceans but there has been criticism of a lack of replication. Here, we report an experiment on the effects of a noxious chemical, sodium hydroxide, applied to one eyestalk of the glass prawn. This caused an immediate escape tail-flick response. It then caused nipping and picking with the chelipeds at the treated eyestalk but much less so at the alternative eyestalk. Prior treatment with benzocaine also caused an immediate tail-flick and directed behaviour, suggesting that this agent is aversive. Subsequently, however, it reduced the directed behaviour caused by caustic soda. We thus demonstrated responses that are consistent with the idea of pain in decapod crustaceans.

## 1. Introduction

Animals often encounter situations in which tissue damage occurs, and that damage might have major negative impacts on fitness [1,2]. However, the early evolution of nociceptors provided a means of detection of such damage and enabled animals to withdraw all or part of their body using a nociceptive reflex [3]. Nociception thus provides a system that should stop the damage in the short-term [4]. In some animals, a second system has evolved, which is called pain. Pain is defined by the International Association for the Study of Pain (IASP) as “An unpleasant sensory and emotional experience associated with, or resembling that associated with, actual or potential tissue damage” [5]. The unpleasant sensory and emotional experience must have some function beyond that of nociceptive reflexes, and it is generally accepted that it provides long-term protection by changing behaviour in ways that prevent further damage and promote recovery [2].

A key problem in the study of possible pain is that a reaction to a noxious stimulus might just be a nociceptive reflex [4]. However, it is widely accepted that humans, and at least some other vertebrates, experience pain following a nociceptive input [2]. Note, however, that absolute certainty about pain experience is not possible [4]. To consider that pain is a possibility requires various criteria to be fulfilled. Various lists of criteria have been constructed, some long [2] and some short [6]. Because the function of pain seems to involve a relatively long-term alteration of behaviour, it has been suggested that an emphasis be put on behaviour that is not easily explained by reflexes [7]. The more that criteria are fulfilled for a particular taxon, the more likely it is that members of the taxon experience pain.

One behavioural criterion for pain is that it should elicit behaviour directed toward the site of the noxious stimulus, or protection of that site by guarding or limiting the use of that part of the body [2,6,7]. For example, we expect to see rubbing, licking, holding, guarding, or limping, which can be too complex and prolonged to be explained by reflexes, and these behaviours have been noted in a broad range of animals [2]. A second expectation of pain is that the reaction to the noxious stimulus be ameliorated by analgesics or local anaesthetics [2,6,7] and, again, these reactions have been noted in a range of animals [2]. Here, we examine these two criteria in a decapod crustacean, the glass prawn, *Palaemon elegans*.

These criteria have received attention in *P. elegans* [8]. Brushing of caustic soda or acetic acid on one antenna caused immediate tail-flicking escape responses, which may be nociceptive reflexes, that were not noted when just sea water was applied. Further, there was a marked increase in grooming of the specific antenna and rubbing that antenna against the side of the tank that was not apparent when sea water was applied. Prior brushing with a local anaesthetic was initially noxious, as suggested by immediate tail-flicking escape responses and, subsequently, more grooming of the antenna was observed. However, in animals in which caustic soda or acetic acid was applied after the local anaesthetic, there was a reduction in grooming and rubbing of the noxiously treated antenna compared to the animals that did not receive the local anaesthetic. These findings were viewed as being consistent with the idea of pain [8]. An attempt to replicate the study in three different species of decapods, however, found no effect of either caustic soda, hydrochloric acid or local anaesthetic on antennal grooming or tail-flick responses [9]. This led other authors to state that there is no reliable evidence for decapods being sensitive to extreme pH and they criticised the lack of replication of this type of study [10]. This call for replication provided the impetus to report on an experiment conducted some years ago. This experiment used one eyestalk of a glass prawn as the site of application of caustic soda to determine if this caused immediate tail-flick escape responses and/or behaviour directed to that eyestalk. We also investigated the effects of prior treatment with a local anaesthetic, benzocaine, on the behaviour of the glass prawn.

## 2. Materials and Methods

### 2.1. Collection and Experimental Treatments

*P. elegans* were collected in hand nets from rock pools during low tide on the shore at Ballywalter, Co Down, Northern Ireland (OS; J 634708), between November 2006 and January 2007. They were immediately transported to Queen’s University Belfast and housed in tanks containing aerated sea water, maintained between 11 °C and 13 °C on a 12 h light/12 h dark photoperiod regime, with seaweed (*Fucus serratus*) present in the tanks. Before each treatment, a prawn was removed using a small net and placed in a glass dish containing seawater, covered with a paper towel to prevent the animal escaping, and transferred to an adjacent observation room. The prawns were randomly assigned (by drawing tokens from a bag) to one of four experimental groups (*n* = 18 per group). Each animal was subject to two sequential treatments.

For the first treatment, the animal was placed into a clean dish containing paper towel dampened with seawater. Then, either seawater or 2% benzocaine solution was applied to a randomly chosen (by coin toss) eyestalk using a small brush. One application was carried out along the eyestalk to the tip, and a separate brush was used for each treatment and for each prawn. An immediate reaction to the treatment in the form of a tail-flick escape response was noted. The prawn was then placed in an observation tank (19.5 × 9 × 9 cm) containing fresh seawater (11–13 °C) to a depth of 3 cm. The observation tank was housed in an observation chamber behind a one-way mirror and the behaviour was recorded for 5 min. The activities recorded were (a) the time taken to first cross a marked line that divided the tank in half, (b) the number of times the prawn crossed the line, (c) the number of tail-flick movements (d) the amount of time the animal spent grooming its treated eyestalk and (e) the amount of time spent grooming the untreated eyestalk. Grooming of the eyestalks consisted of the animal remaining stationary and using its chelipeds to nip and pick at its eyestalks.

For the second treatment, the prawn was removed from the observation tank and placed into a new, clean treatment dish containing a paper towel dampened with seawater. The same eyestalk was then either treated with seawater or 10% NaOH following a similar procedure as before. The prawn was then placed into the observation tank for another 5 min, and the same activities were recorded.

### 2.2. Statistical Methods

*First treatment:* The occurrence of tail flicking immediately following the first treatment was determined using χ^2^ contingency tests. The occurrence of tail flicking in the observation tank was determined using χ^2^ contingency tests. Differences in general activity, as indicated by the time taken to cross the line and the number of line crosses, were ascertained using unpaired *t*-tests. The effects of water or benzocaine on the treated and untreated eyestalks were analysed using two-factor ANOVA (with factor 1 being water or anaesthetic, and factor 2 being the repeated measure of treated eyestalk or untreated eyestalk).

*Second treatment:* The occurrence of tail flicking immediately following the second treatment was determined using χ^2^ contingency tests, in relation to the first and second treatments. Differences in the occurrence of tail flicking in the observation tank were determined using χ^2^ contingency tests. Differences in general activity, as indicated by the time taken to cross the line and the number of line crosses, were ascertained using a two-factor ANOVA (with factor 1 being the first treatment with benzocaine or water, and factor 2 being the second treatment with NaOH or water). The effects on the treated and untreated eyestalks were further analysed using three-factor ANOVA, with treated and untreated eyestalk as repeated measures (factor 1: water or anaesthetic; factor 2: NaOH or water; factor 3: treated or untreated eye as a repeated measure). The grooming of just the treated eyestalk was also analysed using a two-factor ANOVA (factor 1: first treatment, factor 2: second treatment).

### 2.3. Ethical Considerations

This experiment was conducted in 2006/2007, when there was little or no support for the idea of pain in decapod crustaceans. There were no legal restrictions on experiments on this group of animals. However, similar experiments which applied noxious stimuli to animals, published between 2008 and 2017 [4], provided evidence to suggest the idea of pain in decapods. Nevertheless, we kept the numbers of animals low in each experimental group (*n* = 18 per group) and we anticipated that only one group would be subject to an aversive experience. However, the data on benzocaine treatment subsequently suggested that three groups would have an aversive experience, which was more than expected. The results of similar experiments have changed the legal situation for decapods within the United Kingdom, which now recognises that these animals are sentient. Despite this, there has been no change in the UK legal requirements for research on these animals, and thus the experiment is fully compliant with current UK regulations. Nevertheless, we suggest that researchers should take the potential sentience of these animals into account when designing future studies rather than waiting for legal change and refer to guidelines on the use of wild animals [11].

## 3. Results

### 3.1. First Treatment: Effects of Seawater or Anaesthetic

Significantly more animals flicked their tails upon application of anaesthetic compared with seawater (32/36 vs. 0/36; χ^2^_1_ = 57.6; *p* < 0.0001). However, the time taken to first cross the line during the 5 min observation did not differ significantly between treatments (t_70_ = −1.421, *p* = 0.1599) and there was no significant difference in general activity as indicated by the number of line crossings (t_70_ = 1.3, *p* = 0.2). There was no significant difference in the occurrence of tail flicking during the 5-min observation between animals treated with water or anaesthetic (9/36 vs. 7/36; χ^2^_1_ = 0.32; *p* = 0.57).

For eyestalk grooming, there was a significant interaction effect between whether the eyestalk had been treated or not and the nature of that treatment (F_1,70_ = 10.01, *p* < 0.01; Figure 1). This interaction was due to the high level of grooming of the treated eyestalk when the first treatment was benzocaine rather than water. Overall, there was a significant effect of first treatment (F_1,70_ = 13.2, *p* < 0.001; Figure 1), with more grooming occurring when the first treatment was anaesthetic, and grooming was directed more towards the treated eyestalk than the untreated eyestalk (F_1,70_ = 7.96, *p* < 0.01; Figure 1).

### 3.2. Second Treatment: Effects of Seawater and NaOH Following First Treatment

Significantly more animals flicked their tails upon application of sodium hydroxide compared with seawater (28/36 vs. 0/36; χ^2^_1_ = 45.82; *p* < 0.001). However, first treatment (water vs. anaesthetic) did not significantly affect the occurrence of tail flicking upon application of the second treatment (13/36 vs. 15/36; χ^2^_1_ = 0.234; *p* = 0.63).

There was no significant effect of the first treatment on the time taken to first cross the line (F_1,68_ = 0.49, *p* = 0.49). However, the second treatment did have a significant effect (F_1,68_ = 5.68, *p* < 0.05), with prawns that underwent sodium hydroxide treatment taking longer to first cross the line. There was no interaction effect between first and second treatments (F_1,68_ = 1.47, *p* = 0.23). The number of line crosses during the second session was not significantly affected by either the first treatment (F_1,68_ = 0.03, *p* = 0.87) or the second treatment (F_1,68_ = 0.64, *p* = 0.43) and there was no interaction (F_1,68_ = 0.37, *p* = 0.55). There was no significant difference in the occurrence of tail flicking in the observation tank between animals first treated with water or anaesthetic (10/36 vs. 4/36; χ^2^_1_ =3.19, *p* = 0.07). Tail flicking did not differ depending on the nature of the second treatment (χ^2^_3_ = 0.36, *p* = 0.55).

The three-way interaction between the first and second treatments and which eyestalk was groomed was not quite significant (F_1,68_ = 3.75, *p* = 0.06, Figure 2). However, the two-way interaction between whether the antenna was treated or not and the nature of the second treatment was clearly significant (F_1,68_ = 194.36, *p* < 0.001, Figure 2), showing that most grooming was directed towards the treated antenna when the second treatment was NaOH. There was a significant interaction effect between the first and second treatments (F_1,68_ = 4.6, *p* < 0.05, Figure 2) because there was more grooming when the first treatment was water rather than benzocaine and the second treatment was sodium hydroxide rather than water.

When only grooming of the treated eyestalk was analysed, there was a significant effect of first treatment (F_1,68_ = 8.72, *p* < 0.05), with more grooming occurring in the second session when the first treatment was seawater. The second treatment had a significant effect on grooming (F_1,68_ = 604.63, *p* < 0.001), with most grooming following treatment with sodium hydroxide. Importantly, there was a significant interaction effect (F_1,68_ = 8.72, *p* < 0.05) due to the high level of grooming of the eyestalk when the first treatment was water and the second treatment was sodium hydroxide, but there was less grooming if the first treatment was benzocaine.

## 4. Discussion

Brushing the eyestalk with seawater was not noxious, as no animal showed a tail flick at that time. This is in marked contrast to the immediate response to being brushed with benzocaine, following which virtually all animals showed an immediate tail-flick response. Tail flicking is the key escape response in prawns, and, in water, the animal would be propelled backwards. This response to benzocaine was unexpected. However, there are reports that benzocaine is often prepared in an acid solution [12] and topical application may cause stinging or burning sensations in humans [13]. Note, however, that this response to benzocaine was not observed in experiments on three other decapod species [9].

When the animals were returned to the water, there was no effect of the first treatment on general activity, either in terms of time to first crossing a mid-line in the tank or total number of line crossings. Also, there was no effect of the local anaesthetic on tail flicking while the animals were in the water. However, behaviour was directed towards the treated eyestalk when that had been brushed with benzocaine; the animal used its chelipeds to pick and nip at the eyestalk (Figure 1). This demonstrates that the aversive effect of benzocaine persisted after the animal was returned to the seawater. The interaction term shows that the grooming was directed primarily, but not entirely, at the treated antenna. It is not clear why a minor amount of grooming was directed to the untreated eyestalk. The eyestalks are on either side of the head and sufficiently separated to ensure there was no accidental transfer of benzocaine from one side to the other. Thus, it is the behaviour that appears to be misdirected, albeit in very small amounts.

During the application of the second treatment, only animals receiving sodium hydroxide showed tail-flicking escape responses; however, pretreatment with benzocaine failed to ameliorate this response. When placed in the water, those animals that had sodium hydroxide on an eyestalk took longer to initiate movement, but again, this measure was not affected by prior treatment with benzocaine. A previous experiment on *Palaemonetes* sp. found that sodium hydroxide applied to an antenna resulted in a longer time passing before locomotion was initiated, but that study considered the finding to be a false positive as it was not noted in two other species [9]. That it was also found in the present study, however, suggests that it is a true effect. Overall movement and tail flicking while in the water, however, were not affected by any treatment. Nevertheless, there was a clear indication of sodium hydroxide being aversive because there was significantly more grooming of the eyestalk when NaOH was applied as the second treatment. Further, the grooming was directed significantly more at the treated than the untreated eyestalk. There was some amelioration of the response to sodium hydroxide by pretreatment with benzocaine, but, although statistically significant, the effect was not large. Thus, there is some doubt as to the effectiveness of benzocaine as a local anaesthetic when applied to the eye. It is possible that it does not fully block nerve transmission within 5 min of application. However, the lack of eyestalk grooming during the second observation period in animals pretreated with benzocaine and secondly with seawater indicates that the aversive nature of benzocaine is short-lived. The second observation period occurred about 6–11 min after the application of benzocaine. Future experiments on benzocaine or other local anaesthetics should test over different durations to establish the timing of aversive and anaesthetic properties [14].

This experiment shows that glass prawns respond to potentially noxious chemical stimuli both with immediate tail-flicking, which presumably is mediated by a nociceptive reflex [8], and with the later, more complex and prolonged behaviour directed to the site of application. Sometimes, the animals use one cheliped to do this but at times two are used simultaneously. In this case, the movements of the two chelipeds are different and the joints bend in different ways to reach the treated eye. These activities appear to have a major protective function. The eyes are thus vital for the fitness of glass prawns and visual stimuli in the wild readily elicit escape or feeding (pers. obs.). The eyes are situated at the distal end of flexible eyestalks and these eyestalks have important endocrine functions [15], and virtually all aspects of crustacean physiology are affected by eyestalk removal [16]. The regenerative capability of crustaceans does not extend to the eyestalks, suggesting that regeneration of the central nervous system may not be possible [17]. Thus, the ability to detect noxious stimuli that might damage either the eye and/or the stalk would be beneficial to the animal. Hence, the tail-flicking escape responses seen when noxious, potentially damaging chemicals are applied, and the nipping and picking at the eye and stalk, appear to maintain the integrity of this vital organ complex.

The present findings are similar to those shown by Barr et al. [8]) when antennae were subject to different treatments. However, Diggles et al. [10] expressed deep concern that the previous experiment had not been replicated and expressed surprise that directed grooming was included in the review by Birch et al. [18]. We note, however, that Diggles et al. [10] ignored other reports of chemicals causing extreme responses in various decapod species, despite some of these being reviewed in [19]. For example, shore crabs (*Carcinus maenas*) used their claws to scratch at their mouthparts if they had been brushed with acetic acid [20]. Acetic acid brushed on an eye also caused that eye to be held down for longer than a control-treated eye [20]. Further, shore crabs (*Hemigrapsus sanguineus*) reduced their use of a claw if it was injected with formalin and often pressed that claw against the carapace [21]. Crabs also shook and rubbed the injected claw. Some of these animals subsequently autotomised a claw injected with formalin [21] and similar autotomy occurred if the base of a walking leg of *C. maenas* was injected with acetic acid [22]. We have also seen cases of autotomy due to high temperature [23], damage to a flexible distal joint on a leg [24], and electric shock to a basal joint of a walking leg [25]. A further example of behaviour being directed to the site of a noxious stimulus is seen in hermit crabs that groomed their abdomen if an electric shock was applied [26]. Also, brown crabs (*Cancer pagurus*) that had a cheliped twisted off, as in fishery practice, held their remaining claw over the wound during competitive interactions [27].

These observations of behaviour directed to a specific site are consistent with the idea of pain [4,6]. Similar responses to injection of acetic acid have been reported for octopi [28] and fish [29]. Mammals also direct their behaviour to the site of noxious treatment [30,31,32]. However, not all treatments induce the same responses in crustaceans and mammals. For example, capsaicin induces pain and much directed grooming and rubbing in mammals [33] but it has no effect on decapods [20,34]. This is likely due to differences in nociceptor channels seen between taxa [3].

Experiments that examine the possibility of pain in decapods are important for understanding animal welfare. Decapods are fished in the wild, and reared in aquaculture systems, in vast numbers for human consumption, and, until recently, their treatment has not been shaped by welfare concerns [35,36]. The traditional view is that these animals respond to noxious stimuli only by reflex reactions and thus they have no capacity for pain or suffering [19]. However, various experimental approaches have questioned this view because behavioural, physiological and morphological criteria have been fulfilled and thus are consistent with the idea of decapods feeling pain [2,4,6]. Thus, apart from the directed behaviour towards the site of damage, and the modulation of responses by local anaesthetics noted here, we see rapid avoidance learning [37], anxiety [38], trade-offs with other requirements [39,40] and long-term shifts in motivation [26]. These animals will also give up highly valuable resources to escape high-, but not low-intensity noxious stimuli [41]. None of these are easily explained by nociceptive reflexes [4]. Nociceptors have been identified [34], as have suitable brain organisations [42] and physiological changes that indicate stress following noxious stimuli [24,43]. These aspects and many others were reviewed and presented to the UK government and accepted as being sufficient evidence to declare that decapods were sentient [18].

## 5. Conclusions

We show that caustic soda is a noxious stimulus when applied to an eye and it elicits prolonged, directed grooming towards that eye. Benzocaine is also noxious when first applied, but it subsequently reduces directed grooming elicited by caustic soda. These results further fulfil two key criteria for pain.

Whilst we accept that there can be no absolute proof of pain in decapods the evidence that is consistent with sentience is sufficiently widespread within the taxon and of sufficient quality and variety that it is unreasonable to conclude that sentience is not possible [6,44]. Given that sentience is a possibility, protection for this group should be afforded, particularly in the food industry, in which billions of these animals are subject to extreme treatments, as well as in research [18,35].

## Figures and Tables

**Figure 1 animals-14-00364-f001:**
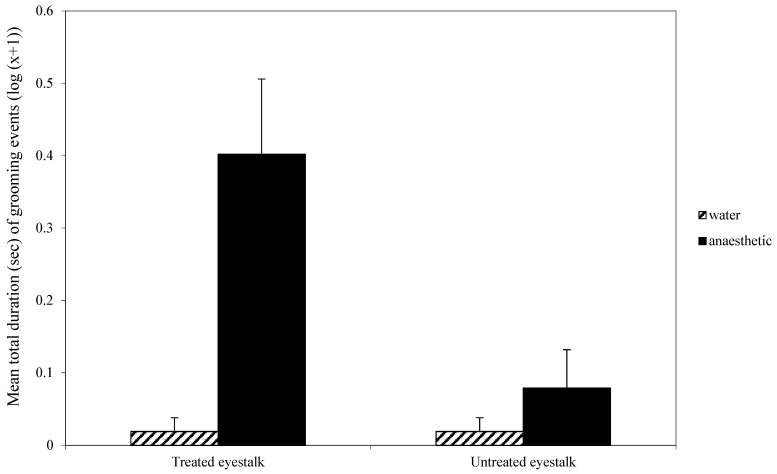
Mean (±SE) total duration (s) (log (x + 1)) of grooming of the treated and untreated eyestalk in the first observation. The low scores for water treatment groups are zero and the small scores shown here were created so those experimental groups may be seen in the figure.

**Figure 2 animals-14-00364-f002:**
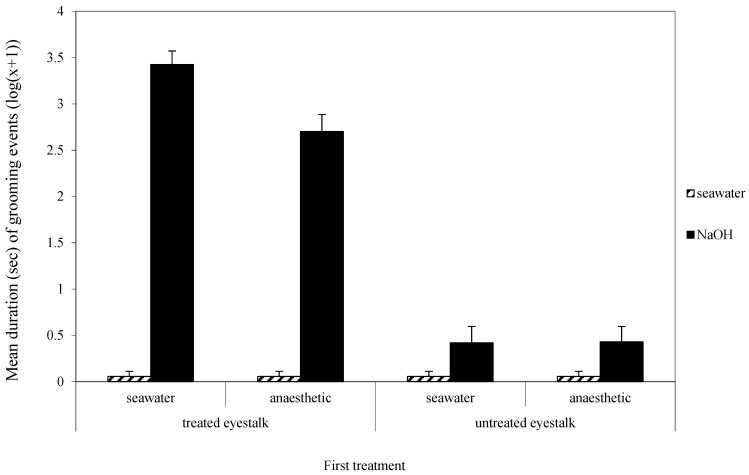
Mean (±SE) total duration (s) (log (x + 1)) of grooming of the treated and untreated eyestalks in the second observation. The low scores for the four water treatment groups are zero and the small scores shown here were created so those experimental groups may be seen in the figure.

## Data Availability

The data presented in this study are available on request from the corresponding author (R.W.E.) The data are not publicly available due to [being in an old program format that will take time to be easily transferred to others].

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
