# Peer review of "The Effects of Caustic Soda and Benzocaine on Directed Grooming to the Eyestalk in the Glass Prawn, Palaemon elegans, Are Consistent with the Idea of Pain in Decapods"

_animals, 2024, doi:10.3390/ani14030364_

Round 1
Reviewer 1 Report
Comments and Suggestions for Authors
This paper is potentially highly significant insofar as it seems to respond to concerns over the replicability of Barr and Elwood's earlier experiments apparently showing that decapods exhibit behaviors consistent with pain. The scientific issue has obvious economic implications given the size of the shrimp market and the fact that Elwood’s and Barr's experiments have raised grave challenges to the commonsense assumption that decapods are not sentient. Their work has given strong reasons to believe several decapod species can feel pain. The data published here seem to buttress the earlier claims and they go some way toward responding to critics' calls for replication. I therefore recommend publication. However, the time at which the experiments were performed should be made more transparent.
1. Does the introduction provide sufficient background and include all relevant references?
The relationship of this work to Barr and Elwood (2008)1 should be clarified. I find potentially misleading the phrases “the current experiment” (line 133) and “the current work” (l. 144), because “current” might be taken to mean “recent.” It appears that the work was done in 2006-7, prior to the 2008 publication. If my reading is correct, the work was done before the criticisms of Key (2016), Diggles (2019), and Diggles et al. (2024). These critics have alleged, as I say, that the authors have not replicated their results. If my interpretation of the timeline is correct, readers might be misled into concluding that the authors are reporting work designed to respond to the critics. However, if the work reported here was done before 2008, as it appears to have been done, this fact should be acknowledged. This fact may affect the reader's interpretation of the extent to which the work responds to the call for replication.
Authors may also wish to note the possibility that some decapod species are more likely to feel pain than others. Line 264 responds to a line of criticism raised by Diggles et al. (2023). Emerging from the work of Puri and Faulkes (2010), who found no nociceptors for extreme pH in three decapod species, Diggles et al. and others have complained that Barr and Elwood have not replicated their 2008 findings. However, two of the 3 species studied by Puri and Faulkes--and hypothesized not to be sentient--were Dendrobrachiata. But nearly all of the species studied by Barr and Elwood have been Pleocyamata. Thus the possibility emerges that pain may be experienced by some but not all decapods. For example, Comstock (2022) points out that pain seems better established in Pleocyamata than in Dendrobrachiata, an idea endorsed by Biffra (2022) and based on judgments expressed in Birch et al. (2021).
The problem just discussed might be addressed at two points.
1. Line 51. "The more that criteria are fulfilled for a particular taxon, the more likely it is that the taxon experiences pain." First, taxa do not experience pain; only individuals feel pain. Second, one cannot generalize from studies on two dozen decapod species to all decapod species.
2. Line 294. "However, various experimental approaches ... and thus are consistent with the idea of decapods feeling pain [2,4,6]." The last clause should read "...and thus are consistent with the idea of at least some decapods feeling pain."
Line 135 refers to ”similar experiments.” The experiments in question are referenced later in the article, but citations early on might help clarify the timeline problem.
2. Minor editing of English language required
Line 215. This sentence seems unclear to me. I think the conjunction “and” should be replaced with a semi-colon. My confusion is that the phrase following “and” provides an illustration of “behavior” referenced in the first clause. However, as written, the phrase following “and” seems to represent a second qualifier which must be added to the first qualifier (that is, “eyestalks brushed with benzocaine”).
Line 226. This sentence seems unclear to me. I think the word “in” should be replaced with “animals.”
Line 298. "leaning" should be "learning"
I am not qualified to evaluate the authors’ statistical analyses or interpretation of the data.
Comments on the Quality of English LanguageThe quality of the English Language is excellent.
Author Response
I have attached the wrong document and cannot change it. The correct document is attached for the other reviewers.

Reviewer 2 Report
Comments and Suggestions for Authors
In my view, this is a valuable and timely contribution to the literature. Minor comments below.
Simple Summary:
**L17-18: “decapod crustaceans” might be too much jargon for the simple summary.
Introduction:
**L34: “nociceptors” could be defined.
**L54-57: The link between pain and directed wound-tending could be unpacked. Why does this go beyond a nociceptive reflex?
**L59: “by analgesics or local anaesthetics BECAUSE THESE AGENTS MODIFY PAIN EXPERIENCE” – or something to again link the criterion to pain.
**L62: As some readers will skim the Intro by reading the first sentence of each para, I’d give the sp. name here (and potentially the criteria too).
**L62: “in this species, BUT FINDINGS HAVE BEEN MIXED” – or something to signal the inconsistent findings that justify this study.
**L63: I think it should be clarified here that tail-flicking is interpreted as a nociceptive reflex, rather than evidence for pain.
**L76: This reference (10) falsely states that its authors have no financial conflicts of interest. By violating basic standards of scientific integrity, I don’t think it should be cited (especially as other papers have highlighted the unsuccessful replication). I leave this to the authors’/editors’ judgement.
Methods:
**L84-85: Not necessarily a problem, but it feels odd to respond to a recent call for replication with data nearly 20 years old. I’m just curious why the authors have this additional dataset, and why they didn’t publish it before.
**L126: Close bracket.
Results:
**L162: “was directed MORE towards the”.
Discussion:
**Given that inconsistent previous results are cited as the primary rationale for this study, how might the authors explain the difference in their findings from the Puri and Faulkes results? This would be relevant to the Discussion.
**L253: Probably just me, but I’m struggling to follow this sentence.
**L266-281: This info feels pertinent to the Introduction.
Conclusions:
**L307-312: I agree that these are relevant conclusions, but I'd also recommend outlining the headline results of this study.
Reviewer 3 Report
Comments and Suggestions for Authors
This paper is topical and confirms previous studies on the subject hence increasing the strength of the evidence on the subject. However, there is an issue with reference to pain (in the title, abstract and line 282). The paper demonstrates evidence of nociception but the writing chooses the phrasing "the idea of pain" rather than sticking with the finding (nociception) and explaining where this finding sits within the wider research on pain in this species. Validating nociception is of value, but the validity of the conclusions, which focus on sentience is questionable.
